# Physical Activity Interventions and Their Effects on Cognitive Function in People with Dementia: A Systematic Review and Meta-Analysis

**DOI:** 10.3390/ijerph18168753

**Published:** 2021-08-19

**Authors:** Maria Isabel Cardona, Adel Afi, Nemanja Lakicevic, Jochen René Thyrian

**Affiliations:** 1Deutsches Zentrum für Neurodegenerative Erkrankungen (DZNE), Site Rostock/Greifswald, Ellernholzstr. 1–2, 17489 Greifswald, Germany; adel.afi@dzne.de (A.A.); rene.thyrian@dzne.de (J.R.T.); 2Sport and Exercise Sciences Research Unit, University of Palermo, 90133 Palermo, Italy; lakinem89@gmail.com; 3Institute for Community Medicine, University Medicine Greifswald, Ellernholzstraße 1–2, 17489 Greifswald, Germany

**Keywords:** physical activity, cognitive function, dementia

## Abstract

Background: Physical activity (PA) has emerged as an alternative nonpharmacological approach to effectively address the effects of dementia. The primary aim was to identify and summarize PA interventions and their effects on cognitive function among persons with dementia (PwD). Methods: A systematic review was conducted with a meta-analysis using different electronic databases, such as PubMed, Embase, APA PsycNET, and the Web of Science. The identified and selected studies were randomized controlled trials (RCTs) that were written in English, published between 2000 and 2020, and implemented among PwD who received a PA intervention and whose cognitive function was measured at baseline and during a follow-up. Results: Twenty-two PA intervention studies met the eligibility criteria and showed a medium-size effect on the cognitive function of PwD, 0.4803 (95% CI = 0.1901–0.7704), with a high percentage of heterogeneity (I^2^ = 86%, *p* ≤ 0.0001). Moreover, this review complements other reviews by including eight studies that have not previously been considered. Overall, studies have methodological limitations. However, six studies implemented in the past five years have shown more robust methodological designs, including larger sample sizes and more comprehensive measurement tools. Conclusion: It is not yet possible to draw a conclusion on the ideal PA intervention for this population due to the high proportion of heterogeneity within the included studies. More emphasis is needed on the intensity of PA monitoring and adherence to such programs.

## 1. Introduction

Recent findings indicate that the population has been rapidly ageing during the last century due to improvements in health care, increase in life expectancy, and decrease in fertility rates [1]. As people age, body organs, tissues, and cells undergo change. Histological studies have shown that ageing affects the central nervous system (CNS) since it experiences neuroanatomical alterations, including an overall reduction in brain activity [2,3]. Therefore, changes and damage in the CNS are worrisome, due to its decisive role in controlling and coordinating essential functions of the body, including cognitive functions [4]. The physiological characteristics of dementia, an umbrella term for multiple neurodegenerative diseases [5], has been linked to the severe degeneration of brain cells and synapses in certain areas of the CNS, including the temporal, parietal and frontal cortices [6]. Damage in these areas manifests itself through memory and learning deficits [6]. In addition, dementia affects emotional regulation, social functioning, and activities of daily living [5]. According to the World Health Organization [7], there are 47 million people with dementia worldwide today, and it is expected that by 2030 this number will rise to 75 million and in 2050 to 135 million. Considering dementia’s impact, researchers have concentrated efforts to minimize the burden associated with this disease by studying dementia risk factors and evidence-based dementia prevention and treatments [8].

The causes of dementia onset are not fully understood, but notably, the mechanism underlying dementia is associated with abnormal protein deposits that coexist with neurovasculature at different stages of the disease, which affect the functioning of the brain [9]. Depending on the type of dementia, different protein accumulations are observed. For instance, alpha-synuclein protein is linked to Lewy body dementia, whereas beta-amyloid and tau proteins are both related to Alzheimer’s disease (AD), the most common form of dementia. Inadequate blood flow can lead to vascular dementia [9]. Other non-modifiable factors linked to dementia include age, sex, inflammation, and comorbidity, and genetic, environmental, and lifestyle factors [10]. Particularly in recent years, substantial epidemiological studies have provided evidence for lifestyle-related risk factors that trigger the development of dementia [11,12,13]. In light of this, the Lancet Commission presented a model describing nine modifiable risk factors (e.g., physical inactivity) that may contribute as much as 35% to the risk of dementia across the lifespan. Thus, by modifying these risk factors, one has a higher chance of preventing or delaying dementia progression [14].

In particular, PA during midlife and late life has been considered a cognitive reserve-enhancing factor associated with a decreased risk of developing dementia [11,12,14]. This is mainly because regular PA improves the strength of cells and tissues to respond to oxidative stress, vascularization, and energy metabolism and also allows neurotropic effects through neurotrophic factor (BDNF) concentrations, which contribute to brain plasticity, memory improvement, neurogenesis, and synaptic plasticity [15]. These processes attenuate for the loss of brain tissue while the brain is ageing [14]. Thus, PA is linked with the concept of increased cognitive reserve, which indicates the brain’s resilience. Persons who present this condition are more likely to cope with nervous system tissue damage without cognitive degeneration [14]. Moreover, the positive effects of PA on cognition appear to be influenced by preventing cardiovascular risk factors (e.g., obesity, hypertension, diabetes) which, at the same time, are linked with greater probability of dementia progression [16]. Additionally, neuroimaging methods add further evidence of the impact of PA on brain activity and cognitive function [16]. For instance, an enlarged level of connection was detected between the default mode network (DMN), which is a control structure widely known to be responsible for introspection and memory retrieval, after PA training [17]. Precisely, animal models of Alzheimer’s disease (AD) illustrate that PA is an effective way to positively modify pathophysiological processes, including β-amyloid (Aβ) burden, tau phosphorylation, and neuronal loss [18].

In this way, PA plays a crucial role in the healthcare system. Including preventive and care strategies for dementia that promote resilience and healthy lifestyles, such as PA, may delay the onset and progression of dementia [14]. PA is understood “as any bodily movement produced by skeletal muscles that require energy expenditure above and beyond resting energy expenditure (one metabolic equivalent = 1 MET) and it can be undertaken in many different ways: walking, cycling, sports and active forms of recreation” [19]. Additionally, PA can be classified into different intensity levels: light (1.6–2.9 MET), moderate (3–5.9 MET) and vigorous (≥6 MET), each of which are based on the subjective intensity perception of an individual. Thus, this classification denotes, through MET values, the energy expenditure and/or the amount of oxygen consumed while sitting or performing a PA [20].

Although the positive effects of exercise on cognition in older adults have been researched, the influence of PA on cognitive function of PwD is still not well understood [16]. Scientific intervention studies have emerged to provide evidence for the efficacy of PA as a cognitive reserve-enhancing factor and to assess its potential in delaying cognitive decline in PwD. In two recent meta-analyses, [21,22] considering evidence up to 2018, one showed that 13 RCTs with 673 subjects diagnosed with AD presented statistically significant improvements in cognition after participating in PA interventions (SMD = 1.12 CI: 0.66~1.59) [21]. The second meta-analysis [22] involved 13 RCTs with 659 subjects with AD and reported that PA had a positive effect on cognitive function among persons with AD (*p* = 0.003). Overall, previous reviews have reported that PA might positively affect the cognition of PwD given its potential to delay cognitive impairment. However, these studies revealed inconclusive results associated with methodological issues and heterogeneity. Such conclusions are in line with other reviews published in recent years [23,24]. For instance, Forbes et al. [25] stated that no clear evidence was found regarding the effects of PA on cognitive activity (95% CI −0.05 to 0.92, *p*-value 0.08; 9 studies, 409 participants) due to considerable heterogeneity (I^2^ value 80%) and deficient quality of the reported evidence.

Therefore, in order to obtain more conclusive results, multiple reviews [21,22,23,24,25] have emphasized that new trials should address methodological barriers by including larger sample sizes [21,22,23,24] and other strategies as follows: providing standardized intervention characteristics [21,24]; providing more information about randomization processes, blinding, attrition rates, and adverse events [25]; conducting different measurements throughout the intervention period [23]; implementing long-term follow-up measures [21,22,23]; using improved and more sensitive cognitive measures [23]; targeting the type of the disease [24]; targeting stage of the disease [22]; separately assessing subjects with Alzheimer’s disease and vascular dementia [24]; including different types of PA [23]; and ensuring that the control group does not perform the same amount of PA as the experimental group [23].

Although the effects of PA on dementia patients’ cognition have been widely studied over the last few years, it remains unclear whether these recommendations have been integrated into the latest trials and whether increasing methodological quality influences the homogeneity of the results obtained, particularly since the last existing meta-analyses [21,22] mostly included studies conducted before 2015. Therefore, we wanted to provide an update concerning the latest occurrences regarding the new RCTs implemented in the field.

## 2. Objective

### 2.1. Primary Objective

To identify the effects of PA interventions on cognitive function in individuals diagnosed with dementia compared to those in the control group.

### 2.2. Secondary Objective

To recognize if recent PA interventions address methodological barriers reported in previous reviews and provide clearer conclusions about the effects of PA on cognition in PwD.

## 3. Materials and Methods

### 3.1. Methodological Approach

To have clear guidance while conducting the systematic review, we followed the Preferred Reporting Items for Systematic Reviews and Meta-Analyses (PRISMA) set of items to report systematic reviews and meta-analyses [26].

### 3.2. Criteria for Inclusion

Studies were considered eligible if they were RCTs in which participants were randomly assigned to a PA group or a control group. The exercise group required implementing a PA program, including strength, aerobic, and balance exercises, as well as interventions combining physical and cognitive exercises for improving the cognitive performance in PwD. In addition, there was no time restriction; interventions could cover any length and duration. In contrast, the control group consisted of usual care, social activities, or handicrafts. Moreover, participants had to be diagnosed utilizing valid criteria, including the Mini-Mental State Examination [MMSE] (cut-off scores for MCI ≤ 24, ≤21, and ≤19); the Montreal Cognitive Assessment [MoCA] (cut-off scores for MCI were ≤25, ≤24) [27]; the Diagnostic and Statistical Manual of Mental Disorders [28]; the National Institute of Neurological and Communicative Disorders and Stroke; and the Alzheimer’s Disease and Related Disorders Association [29], or ICD-10 [30]. All forms of dementia diagnosis and severity were included. Trials measured cognitive function with a neuropsychological or cognitive test at baseline and follow-up. Finally, studies that were published in English between 2000 and 2020 were included. The primary outcome involved individuals with dementia and addressed their cognitive function.

### 3.3. Criteria for Exclusion

Studies excluded were pilot RCTs, systematic reviews, meta-analyses, study protocols, and conference publications. Studies were also excluded if the intervention was targeted at participants with mild cognitive impairment, PA training was implemented without assessing cognition, or multimodal interventions were conducted without a PA component.

### 3.4. Search Strategy

A search strategy was conducted on two different occasions (January and May 2020). Moreover, the search was performed for RCTs studying the efficacy of PA in four different databases: PubMed, Embase, APA PsycNET, and the Web of Science from the 1st of January 2000 until May 2020. To obtain the search results, we combined relevant English keywords such as physical activity, dementia, cognition, and RCTs (see Appendix A for full electronic search). Furthermore, in May 2020, we performed an additional hand search screening of pertinent studies’ bibliographies to identify articles that the initial search strategy did not recognize. Two independent reviewers (MC and AA) conducted this search, screened initial titles and abstracts, and retrieved the full text of potential papers. A third author was consulted when discrepancies emerged.

### 3.5. Study Selection

Initially, titles and abstracts were imported to EndNote; then, they were screened, and duplications or studies that were determined as irrelevant were omitted. Subsequently, full-text articles from the possible pertinent studies were screened in detail. At this point, studies that met the inclusion criteria were included. All data were independently scanned and selected by two reviewers. In the case of discrepancies, a third evaluator was consulted. This process for selecting studies is shown in the PRISMA flow diagram in Figure 1 [26].

### 3.6. Data Extraction

A data extraction sheet was designed to provide accurate data on PA programs among PwD. Information regarding participants, dementia severity at baseline according to the Mini-Mental State Examination (MMSE), the intervention group, the control group, length-frequency-duration, PA intensity, cognitive assessment, follow-up, adherence rate, and the impact on cognition was documented in a tabular form. Moreover, the means and standard deviations were extracted from global cognition measurements at baseline and at the end of the study. A *t*-test was used to determine statistical significance for global cognition. In some studies, this data was not available. Hence, corresponding authors were contacted, and those who did not respond were not considered for inclusion in the analyses.

### 3.7. Synthesis of Results

A random-effect meta-analysis was carried out to evaluate global cognition outcomes in PwD due to heterogeneity among the studies. Furthermore, considering that studies reported continuous outcomes, assessed at baseline and follow-up, we pooled means and standard deviations.

Moreover, a qualitative synthesis of the results was performed to understand what kind of PA components might be most effective in improving cognitive function among PwD. This summary is articulated based on the content characteristics and methodological aspects of PA interventions and their effects on the cognition of PwD.

### 3.8. Methodological Quality Assessment

Two independent reviewers (MC and AA) assessed the risk of bias of the included trials using the Effective Public Health Practice Project (EPHPP) Quality Assessment Tool for Quantitative Studies [31], considering sections A to F (A. selection bias; B. study design; C. confounders; D. blinding; E. data collection method; and F. withdrawals and dropouts). According to the instrument dictionary, each of these components were rated using the codes “strong”, “moderate” and “weak”. An overall strong score was given when there were no weak ratings, a moderate overall score when there was one weak rating, and a weak overall score when there were two or more weak ratings.

Additionally, to provide a more detailed overview of the methodological aspects of the studies, we completed a systematic assessment based on previous review recommendations [21,22,23,24]. We included aspects such as the application of comprehensive cognitive measures [23], measurements throughout the intervention period [23], long-term follow-up [21,22,23], target dementia type [24], target dementia stage [24], and the provision of clear and available information on PA dose responses [21,24]. For this assessment, we counted and reported the number of recommendations fully incorporated into each study. An additional file shows more in detail previous reviews recommendations on methodological aspects (see Appendix A).

## 4. Results

### 4.1. Study Selection

After conducting the electronic search in different databases using the established search terms, 5204 results were yielded. To this total amount, four articles from the hand search thought to be relevant were added. After screening titles and abstracts and removing duplicates, 4884 studies were excluded from further analysis. The remaining 324 studies were selected for full-text screening. Of those, 302 articles did not meet the inclusion criteria. Therefore, 22 studies were included in the present systematic review. Figure 1 illustrates the study selection process according to the PRISMA flow diagram [32].

### 4.2. Participants at Baseline

This review presented studies with 2371 participants diagnosed with dementia (see Table 1). The included studies had sample sizes that ranged from 19 to 494 participants (M = 102.57, SD = 104.703). For dementia type, nearly half of the sample (47.8%) included subjects with AD combined with other dementia types, such as mixed dementia and vascular dementia [33,34,35,36,37,38,39,40,41,42,43]. Moreover, 30.4% included participants with AD [44,45,46,47,48,49,50], and 21.7% involved persons with undefined dementia [51,52,53,54]. Regarding dementia severity, the RCTs presented MMSE scores at baseline that ranged from 12.0 to 24.0 (MS = 17.1, SD = 3.6). Thus, the majority of studies included participants with moderate dementia (47.8%) [33,34,36,38,40,45,47,48,51,52,53], followed by mild dementia (34.8%) [35,39,42,43,46,50,54] and severe dementia (17.4%) [37,41,44,49]. Finally, 60.9% of participants lived in institutions [33,36,37,38,40,41,43,44,45,49,51,52,53,54,55,56], while 39.1% resided in community dwellings [34,35,39,42,46,47,48,50].

### 4.3. Assessment Methods of Cognitive Function

Concerning cognitive assessments, a high proportion of studies [33,34,35,36,37,38,39,40,41,42,43,44,45,46,47,48,50,51,54] measured global cognitive function using the MMSE and/or the Alzheimer’s disease Assessment Scale-Cognitive Subscale (ADAS-Cog). Approximately 39.1% had only implemented the tests mentioned above [34,36,39,40,41,42,44,49,50]. In addition, 60.9% of RCTs added further tests that measured not only global cognition but also other cognitive domains [33,35,36,37,38,43,45,46,47,48,51,52,53,54], including memory [35,43,47,51,53], attention and concentration [35,46,54], language [38,46,47], visuospatial abilities [33,45,48], and executive functions [37]. An additional file shows more in detail implemented measurement tools and cognitive domains measured by the included studies (see Appendix A).

### 4.4. Measurement Periods

Generally, 43.5% [33,39,40,42,44,45,46,49,50] of the trials only conducted pretest and posttest measurements. However, 30.4% [37,41,47,48,51,54,56,57] of all included studies carried out at least one additional measurement during the intervention time. For instance, Öhman et al. [48] performed two measures during the program time, at 3 and 6 months, and Cancela et al. [51] executed 4 measurements at 3, 6, 9, and 12 months. Moreover, 26% of the total sample undertook long-term follow-ups [34,35,36,38,43,52,53]. In particular, Cheng et al. [36] performed follow-ups at 6 and 9 months after the intervention was finished. Likewise, Miu et al. [34] performed follow-ups at 6, 9, and 12 months post-training.

### 4.5. PA Interventions

Regarding control groups, 43.5% [37,40,42,46,48,49,50,52,53,54] received usual care, 26.1% [33,34,39,43,47,52,53] experienced social visits, 21.7% [36,38,44,45,51] performed recreational and handicraft activities, 4.3% [35] received relaxation and flexibility exercises, and 4.3% [41] had daily one-and-one conversations with a therapist. For the experimental groups exposed to PA training, the following characteristics were found regarding PA modality, frequencies and intensities.

### 4.6. PA Modality

More than half (60.9%) [36,37,38,40,41,42,43,46,47,49,50,54] of the interventions implemented combined different types of PA training, including mainly aerobic and strength exercises. For instance, some studies [37,38,41,43,47,50] involved activities such as walking combined with balance and strength seated exercises concentrated on the upper and lower extremities and torso. In addition, two [36,56] interventions implemented Tai-Chi exercises, which involved training for aerobic capacity, muscular strength, and balance. In contrast, 25% [33,34,39,44,51,53] of the studies implemented only aerobic training. Thus, different aerobic activities were carried out, such as cycling [34,39,51]; walking [34,44,53]; and light aerobic exertion of the joints and large muscle groups, accompanied by music [33]. One study [52] included hand-motor training, and 13% of the trials [35,45,48] combined cognitive and PA training, including cognitive and aerobic bicycle training [35]; aerobic, balance, strength and dual tasking training [48]; and cognitive stimulation in addition to stretching and lower-limb aerobic exercises [45].

### 4.7. PA Duration, Frequency, and Total Length

The session duration of all the included interventions ranged between fifteen and one hundred and twenty minutes (M = 48.91, SD = 28.0). Thus, 78.3% [33,34,35,36,37,38,39,41,43,44,45,46,48,49,50,52,53] of the interventions lasted between thirty and sixty minutes, 13% [40,42,47] lasted longer than sixty minutes, and just 8.7% [51,56] provided less than thirty minutes of PA training sessions. Moreover, the frequency per week ranged from two to seven times (M = 3.71, SD = 1.52). Consequently, 39.1% [33,35,36,37,39,46,49,54] reported PA training three times a week, 21.7% [38,40,45,50,52,53] five times a week, and 17.4% [34,42,48] 2 times per week. For the total length of PA interventions, 56.5% [34,36,37,38,39,40,41,42,44,45,46,49,50] of PA programs lasted between three and six months, 26.1% [33,35,43,47,52,53] lasted less than three months, 13% [48,54] lasted seven to twelve months, and only one [51] PA program included an intervention longer than twelve months.

### 4.8. PA Intensity

The included studies presented varied intensities of PA. The majority (30.4%) [44,45,47,49,50,54] considered moderate PA trainings, followed by light PA (26.1%) [33,35,37,51,52,53] and moderate to vigorous PA (21.7%) [38,39,42,43,46]. Some studies [43,45] reported perceived exertion to indicate intensity; in particular, Kim et al. [45] reported light to moderate intensities according to the Borg scale scores (11–13 points), and Toots et al. [38] indicated intensities based on individual degrees of functional deficit. Six studies [35,39,42,45,46] reported maximum heart rate levels. Thus, Karssemeijer et al. [35] and Kim et al. [45] used moderate intensities reflected in maximum heart rates of 65–75%, 40–60% and 30 to 60%, respectively. In contrast, Lamb et al. [42], Hoffmann et al. [46] and Yang et al. [39] described PA intensities of 70–80% of the maximal heart rate from moderate to vigorous.

### 4.9. Adherence Rate

Only 65.2% [34,35,36,37,38,42,43,44,45,46,47,48,51,54] of the RCTs reported adherence rate. From this portion, the majority (56.5%) [34,35,36,38,42,43,44,45,46,47,48,54] indicated having a “good” adherence rate. This positive rate ranged from 59% to 93% (M = 78.06, SD = 11.39). The remaining studies (8.7%) [37,51] rated themselves as having “bad” adherence rates, and no percentages were reported.

### 4.10. Primary Outcome: Effects of PA on the Cognitive Function of PwD

A summary of global cognition outcomes based on the MMSE results (see Table 2) showed that the majority of the studies displayed a significant MMSE mean difference compared to the control group [34,36,39,40,41,42,43,44,49,50,52,53,54].

### 4.11. Meta-Analysis

A meta-analysis was carried out considering sixteen RCTs [34,36,37,39,40,41,42,43,44,46,48,49,50,52,53,54]. The remaining studies did not present data in their publications for the mean outcome indicating global cognition based on the MMSE. The meta-analysis found that PA interventions had a medium-size effect on the cognitive function of PwD of 0.4803 (95% CI = 0.1901–0.7704). Heterogeneity between studies was statistically significant (I^2^ = 86%, *p* ≤ 0.0001) (see Figure 2). To assess publication bias between trials, a funnel plot was carried out (see Figure 3). Evidence was found to be skewed or asymmetric; thus, there was publication bias among the sixteen studies.

### 4.12. Quality Assessment

Each study’s quality was assessed to avoid the risk of bias and provide consistent results. According to the Effective Public Health Practice Project (EPHPP) Quality Assessment Tool for Quantitative Studies (see Table 3), most of the studies presented an overall quality score between strong and moderate. However, in some cases, weak scores were associated with small sample sizes [39,41,44,45,47,49], lacking withdrawal and dropout reports [33,39,40]. In contrast, some studies showed stronger quality scores associated with the selection of bias since they included larger sample sizes (>100 participants) [35,36,38,42,43,46,48,51].

According to the systematic assessment results based on previous review recommendations [21,22,23,24] (see Table 4), the majority of studies included between three and seven recommendations out of nine (Mode = 5). Studies included methodological recommendations associated with implementing a more comprehensive cognitive measures tool that not only assessed global cognition [33,35,36,37,38,43,45,46,47,48,51,52,53,54] but also included measurements throughout the intervention period [37,41,42,47,48,51,54], long-term follow-ups [34,35,36,38,43,52,53], targeted dementia type [44,45,46,47,48,49,50], and targeted dementia stage [44,46,54]. All of the studies provided detailed information on PA type, duration, and frequency. In contrast, the majority of the studies did not provide a clear definition or description of PA intensity [33,34,36,40,41,44,47,48,50,52,53].

## 5. Discussion

This systematic review identified twenty-two RCTs aiming to test the effect of PA on the cognition of PwD. It provides methodologically sounder designs and new results than other studies conducted in recent years. Thus, it adds evidence to other reviews by including new RCTs [35,36,38,45,46,48] that have not previously been included. Overall, the meta-analysis found that PA interventions had a medium-size effect on the cognitive function of PwD. This indicates general positive effects of PA on cognition in PwD. However, the included trials presented a high percentage of heterogeneity (I^2^ = 86%, *p* ≤ 0.0001) as they showed differences in the number of participants, intervention settings, cognitive measurement tools, follow-up periods, PA dose-responses, and reported outcomes. Therefore, these differences between the studies’ methodologies limited the possibility of solid conclusions about the effects of PA on the cognition of PwD. These findings were consistent with a review by Forbes et al. [25], which showed considerable heterogeneity (I^2^ value 80%) and thus inconclusive results.

Even though these results resemble those obtained by Forbes et al. [25], if we look at the newly added trials implemented in recent years, a slight difference is revealed. In particular, six recent trials [35,38,42,43,46,48] showed more powerful designs since they included larger sample sizes. In this way, this finding enables us to see small developments and progress in this particular field of research, including more solid methodological designs and higher statistical power in the most recent studies. Therefore, the results presented in these trials might lead to more precise conclusions about the effects of PA on the cognition of PwD.

Various features of PA interventions could play a crucial role in mediating the effects of PA on cognition, such as PA modalities, dose responses, and intensity. Based on the type of exercise and intensity, changes in the brain’s structure have been obtained [58].

For instance, interventions implemented three types of *PA modalities*: (1) only PA training (cardiovascular or strengthening), (2) combined PA training (cardiovascular and strengthening), and (3) combined PA with cognitive training. According to Bossers et al. [36], combining aerobic PA with resistant training led to improvements in executive functions and memory functions. Thus, the study recommended combining both modalities to stimulate cognitive improvements in both. Öhman et al. [41] attributed improvements in executive functions to dual-tasking (e.g., talking while walking, singing while dancing) and other combined PAs performed at home (strength, balance, and endurance exercises). This study suggested that combined training may enhance the frontal lobe, which is the brain area in charge of executive functions. These results are consistent with one systematic review implemented by Lauenroth et al. [51], who claimed that multimodal PA interventions that consider cardiovascular training combined with resistance training and cognitive tasks resulted in better and more significant outcomes than individual PA training. Moreover, this type of intervention has contributed to improving frontal cognitive functions, global cognition, working memory, episodic memory, executive function, and processing speed [59].

Additionally, studies presented variations in their *PA dose responses*. Their session durations ranged between fifteen and one hundred and twenty minutes; their frequency per week fluctuated between two and seven times, and their total length took between less than three months and longer than twelve months. Particularly, studies in which longer periods of PA were undertaken were more likely to display positive effects. For example, Öhmann et al. [48] implemented a 12-month PA program, which led to positive effects on executive functions among community-dwelling PwD. Moreover, Hoffmann et al. [46] affirmed that PA seems to affect executive function (mental speed and attention) when implemented for at least six months. In the same way, Toot et al. [38] confirmed that for cognition effects, the interventions’ duration seems to play a decisive role. Thus, a four-month program was not enough time to induce cognitive changes. Likewise, Kassermeijer et al. [35] did not show significant effects due to exergaming training. According to the authors, a possible explanation for these results was that people probably needed more time to master the challenges from this type of program, and this RCT implemented a short intervention period of 12 weeks. These results were aligned with one meta-analysis outcome [60] and with one study, which proved that six to twelve months of PA increased cognitive scores and affected brain structure [58].

Regarding *PA intensities* of trial interventions, the majority implemented moderate, followed by light and vigorous PA intensities. However, there is no consensus among studies as to which intensity level might be ideal. Karssemeijer et al. [35] stated no significant effects on executive functions, working memory or episodic memory after implementing a light intensity combined cognitive and PA intervention among community-dwelling persons with mild dementia. In contrast, The Dementia and PA trial [42], which had the largest sample size of the included studies, applied a moderate-to-high intensity PA program; however, these PA intensities also did not result in positive outcomes. Thus, Lamb et al. [42] specified that an exercise program of moderate-to-high intensity improved physical fitness but did not slow cognitive deterioration. Furthermore, participants who took part in the PA arm and had a high intervention attendance displayed worse cognitive decline than the control group. Therefore, according to these authors, there is a possibility that PA may have worsened cognitive impairment. In particular, these negative effects were associated with inflammation and inadequate oxygen supply to certain cortical areas. In this way, this study suggested that high-intensity aerobic and strength exercise should not be used as a method for addressing cognitive deterioration, and future research should examine other forms of PA among dementia patients. Likewise, Toots et al. [38] indicated that high-intensity training did not result in significant differences in global cognition or executive functions. These results are in line with one systematic review [60] that specified that PwD are fragile patients, and excessive and vigorous intensities of PA should be avoided to prevent other health complications. Moreover, monitoring a steady heart rate of 60% of the maximum heart rate might prevent excess complications and burden among patients. In addition, this range might be enough to activate neurobiological responses that benefit the brain functioning of PwD [60]. Further research is needed to clarify the role of intensity in mediating PA effects.

Another relevant aspect for effective PA interventions described in trials was *PA engagement and adherence to* programs due to high numbers of withdrawals in the trials. Thus, studies stated that bad adherence to their program was associated with a lack of motivation [37,49], low emphasis on PA in geriatric facilities, and a lack of knowledge regarding the benefits of PA [51]. Moreover, a high number of persons declined to participate in one study due to a lack of attractiveness of PA, particularly women [42]. Furthermore, one RCT stated that only one specific segment of institutionalized patients joined the study because they were already motivated to perform PA [43]. A current review showed that for healthy adults aged 80 years and older, it was necessary to initiate and adhere to PA to identify its health benefits, overcome physical-activity-associated fear, recognize and prioritize individual PA preferences, receive social support, and minimize environmental barriers [61]. However, considering that PwD present low functional activity and cognitive functioning, it is probable that variables mediating their PA engagement are different compared to those reported by healthy adults [62]. Recent literature lacks evidence on PA participation and adherence-related factors in PwD [63].

Thus, it can be observed that different factors, such as *PA modalities*, *dose responses*, *intensities*, and *engagement and adherence*, play an important role in facilitating effects on cognition in PwD. However, due to the variety of methodologies, contents and results reported in the included studies, the effects of PA on the cognition of PwD remain unclear. Additional evidence is needed, particularly concerning ideal PA modalities, dose-response intensity, and adherence.

Based on the findings from the most recent studies exploring the effects of PA on cognition of PwD, these might be some implications to consider for future research and policy. However, it is essential to consider them with caution, as the reported studies still present certain limitations:Alternative forms of exercise need to be explored for PwD. For example, additional exercises designed to improve functional activity, a variable that has been proven to be influenced by PA among PwD, are needed [42];Exergaming combined with cognitive training is a method that promotes participants’ initiation and adherence to PA through the innovative combination of technology and exercising [35];Engaging in long-term, individualized, home-based training may have some effect on the executive functions of PwD [48];Future programs should also examine the individual characteristics of participants (type and severity of dementia), as they may influence the effects of PA on cognition. It is also essential to examine who may benefit the most from PA [35].

### Limitations

This review aimed to identify current studies and update the scientific evidence on the effects of PA on cognition; it included extensive eligibility criteria. For example, it included participants of all types and severity of dementia. These criteria, therefore, contributed to a high proportion of heterogeneity within the study. Likewise, through the funnel plot, an asymmetric plot was observed, which represented publication bias. Additionally, there was incomplete retrieval regarding effects across all the included studies due to missing data and a lack of responses from the authors. This may limit the quality of the evidence and, thus, should be considered when discussing the results. We interpreted our results carefully to avoid over- or under-estimating the scientific evidence of the methodologically weak RCTs.

## 6. Conclusions

The evidence for the benefits of PA for PwD remains unclear despite the fact that there is increased research activity within the studies identified in this review Furthermore, the selected studies contained stronger methodological aspects compared to reviews conducted in previous years. In addition, considering that certain prerequisites may affect PA programs, further research is needed. In particular, ideal PA modalities, duration, adherence to interventions, and exercise intensity monitoring should be considered.

## Figures and Tables

**Figure 1 ijerph-18-08753-f001:**
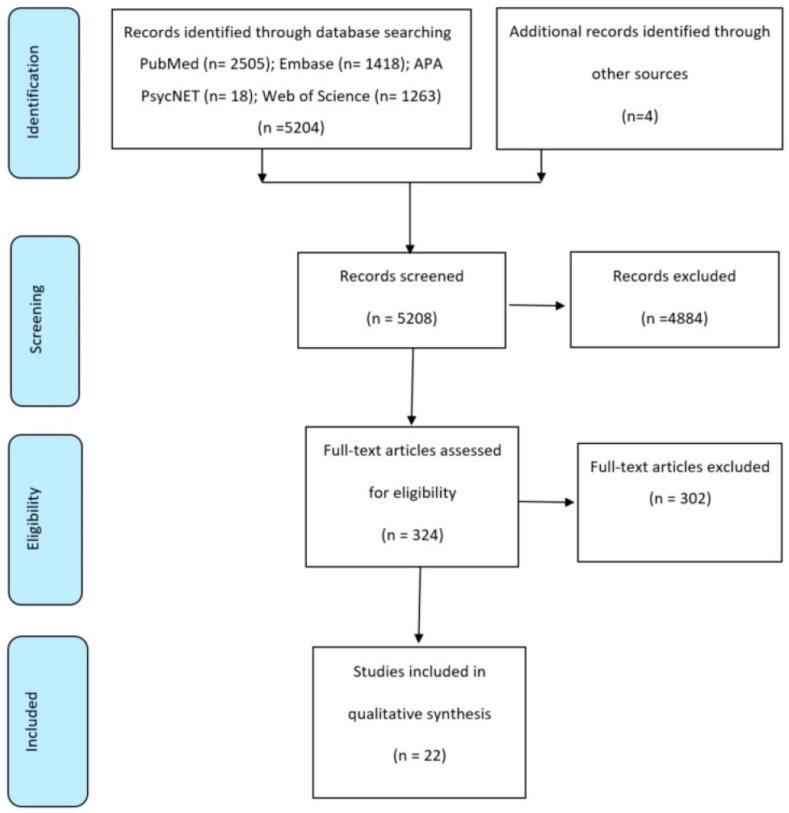
PRISMA flow diagram for trials included and excluded from the systematic review [15].

**Figure 2 ijerph-18-08753-f002:**
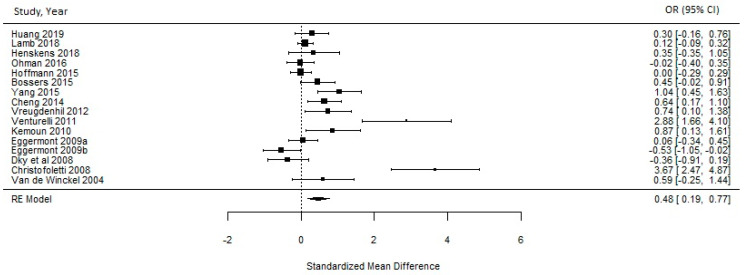
Effects of PA on the cognitive function of PwD.

**Figure 3 ijerph-18-08753-f003:**
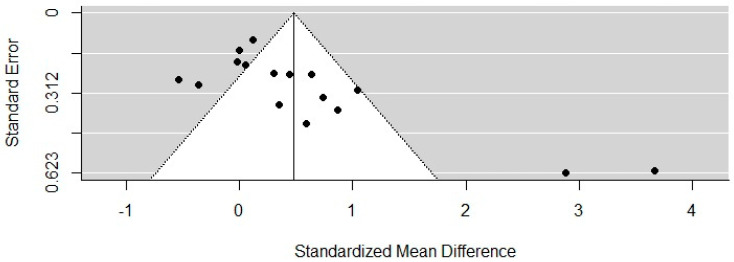
A funnel plot of the included studies.

**Table 1 ijerph-18-08753-t001:** Evidence from included RCTs describing PA interventions.

Study	Participants	IG (n)	CG (n)	Dementia Severity Baseline MMSE	IG	CG	Length Frequency Duration	PA Modality	PA Intensity	Cognitive Assessment	Follow up	Adherence Rate	PA Impact on Cognition
Karssemeijer et al., (2019) [35]	115/AD, VaD, MD/MMSE score ≥ 17/community-dwelling	IG1: 38 IG2: 38	39	22.9 Mild dementia	IG1: cognitive and aerobic bicycle training IG2: Cycling on a stationary bike	Relaxation and flexibility exercises	30–50 min, 3× week, 12 weeks	Combined cognitive and PA training Only aerobic training	Light intensity IG: 41.8% and 43.5% of maximal HR	MMSE; TMT Part B; SCWT; Letter Fluency; Rule Shift Cards Test; WAIS-III (Digit Span); WMS-III (Spatial Span); LLT-R	12 and 24 weeks	Good adherence (85.4%)	−
Huang et al., (2019) [54]	80/UD/MMSE score not specified/care facility	40	40	20.76 Mild dementia	Tai-Chi exercises	Routine treatments and personalized daily care	20 min, 3× week, 10 months	Combined PA training	Moderate exercise intensity	MMSE; MoCA; WHO-UCLA-AVLT; TMT	5 and 10 months	Good Adherence (%NR)	+*f
Lamb et al., (2018) [42]	494/AD, MD, VaD, UD/MMSE score ≥ 10/community-dwelling	329	165	21.8 Mild dementia	Supervised Gym program: Static cycling, arms and legs strength training Unsupervised prescribed Home program	Usual Care	60–90 min, 2× week, 4 months. Plus 60 min, weekly PA at home 150 min each week	Combined PA training	Vigorous intensity	ADAS- Cog	12 months	Good adherence (65%)	−
Henskens et al., (2018) [37]	87/AD, VaD, Mixed VaD and AD, UD/ MMSE score < 24/care facility	IG1: 21 IG2: 22 IG3: 22	22	12.27 Severe dementia	IG1: ADLs training IG2: Multicomponent PA training: Strength seated upper, lower extremities and torso exercises. Outdoor walking IG3: multicomponent PA and ADLs training	Care as usual and social activity intervention	30–45 min, 3× week, 6 months	Combined PA training	Progressive increase intensity	MMSE; SIB-S; GIT (fluency subtest); WAIS (digit Span Task Backward); go-no-go test; FAB (conflicting instructions test)	3 and 6 months	Poor adherence (%NR)	−
Toots et al., (2017) [38]	186/AD, VaD, MD; UD/MMSE score ≥ 10/care facility	93	93	15.0 Moderate dementia	Exercise program for limb strength, balance, and mobility	Structured activities (e.g., singing, reading)	45 min, 5 sessions per two-week period, 4 months	Combined PA training	Vigorous intensity	ADAS- Cog; MMSE; and VF	4 and 7 months	Good adherence (71.5%)	−
Öhman et al., (2016) [48]	210/AD/MMSE score not specified/community-dwelling	IG1:70 IG2:70	70	18.0 Moderate dementia	IG1: Home-base with physiotherapist supervision IG2: Group-based in a day care centre Both implemented aerobic, balance, strength and dual-tasking training	Usual community care	60 min, 2× week, 12 months	Combined cognitive and PA training	NR	CDT; CDR, and MMSE	3, 6, and 12 months	Good adherence (81%)	+ * a
Kim et al., (2016) [45]	38/AD/MMSE score ≤ 20/care facility	19	19	14.8 Moderate dementia	Multicomponent intervention + stretching, lower-limb aerobic exercises using TERASUERUGO	Multicomponent intervention: art and social activities	60 min, 5× week, 6 months	Combined cognitive and PA training	Moderate intensity 40–60% of the maximum HR	ADAS- Cog, MMSE, and CDT	6 months	Good adherence (100%)	−
Cancela et al., (2016) [51]	189/UD //MMSE score not specified/care facility	73	116	15.05 Moderate dementia	Cycling sessions	Recreational activities (e.g., card-playing, craftwork)	15 min, daily, 15 months	Only aerobic training	Light intensity	MMSE; FOME	3, 6, 9, 12, and 15 months	Poor adherence (%NR)	+
Hoffmann et al., (2015) [46]	200/AD/MMSE score ≥ 19/community-dwelling	107	93	24.0 Mild dementia	Strength training of lower extremities and exercises in ergometer bicycle, cross trainer, and treadmill	Treatment as usual	60 min, 3× week, 16 weeks	Combined PA training	Moderate to vigorous intensity. 70–80% of maximal HR	SDMT; ADAS-Cog; SCWT; incongruent score; verbal fluency; MMSE	16 weeks	Good adherence (84%)	−
Bossers et al., (2015) [43]	109/AD, VaD, MD, UD// MMSE score ≥ 9 and ≤23/Care facility	IG1: 37 IG2: 36	36	15.6 Mild dementia	IG1: Two strengthening and two walking sessions per week IG2: Four walking sessions per week	Social visits	30 min, 36 individual sessions, 9 weeks	Combined PA training Only aerobic training	Moderate to vigorous intensity	MMSE; WMS-R; RBMT; SCWT; animals and professions	9 and 18 weeks	Good adherence (89.2%)	+ * ae
Yang et al., (2015) [39]	50/AD, VaD/MMSE score ≥ 10 and ≤24 /community-dwelling	25	25	20.66 Mild dementia	Cycling training	Health education	40 min, 3× week, 3 months	Only aerobic training	Vigorous intensity 70% of maximal HR	MMSE; ADAS-Cog	3 months	NR	+
Cheng et al., (2014) [36]	110/AD, VaD, UD/MMSE score ≥ 10 and ≤24/Care facility	IG1:36 IG2: 39	35	18.8 Moderate dementia	IG1: Cognitive stimulation-board game Mahjong IG2: 12-Form Yang style Tai-Chi	Simple handcrafts	60 min, 3× week, 3 months	Combined PA training	NR	MMSE; Forward and backward digit sequence/digit span; delayed recall Categorical verbal fluency	3, 6, and 9 months	Good adherence (%NR)	+
Vreugdenhil, et al., (2012) [50]	40/AD/MMSE score ≥ 10 and ≤28/community-dwelling	20	20	22.0 Mild dementia	Aerobic walking, strengthening and balance training	Usual treatment	30 min, 5× week, 4 months	Combined PA training	Moderate intensity	ADAS- Cog, MMSE	4 months	NR	+
Venturelli, et al., (2011) [44]	21/AD// MMSE score ≥ 5 and ≤15/Care facility	12	11	12.5 Severe dementia	Walking program	Daily organized activities (e.g., bingo, music therapy)	30 min, 4× week, 24 weeks	Only aerobic training	Moderate intensity	MMSE	24 weeks	Good adherence (93.4%)	−
Kemoun et al., (2010) [49]	31/AD/MMSE score ≤ 23 /care facility	16	15	12.8 Severe dementia	Articular mobilization and muscle stimulation trough walking, equilibrium, stamina, and dancing	No PA participation	60 min, 3× week, 15 weeks	Combined PA training	Moderate intensity 60% and 70% of maximal HR	French RECF	15 weeks	NR	+ *
Steinberg et al., (2009) [47]	27/AD/MMSE score ≥ 10/community-dwelling	14	13	17.7 Moderate dementia	Home-based program, caregivers instructed during visits on daily walking, strength training of major muscle groups, balance and flexibility	Home safety assessment	120 min per visit, 3 visits, 12 weeks	Combined PA training	Moderate intensity	MMSE, BNT, HVLT	6 and 12 weeks	Good adherence (59%)	+
Eggermont et al., (2009a) [53]	97/UD/MMSE score ≤ 10 and ≥24/care facility	51	46	17.7 Moderate dementia	Walking program	Received social visits	30 min, 5× week, 6 weeks	Only aerobic training	Self-selected speed	RBMT (face and picture recognition test); eight words test; digit span from the WMS-R; category fluency and letter fluency	6 and 12 weeks	NR	−
Eggermont et al., (2009b) [52]	61/UD/MMSE score ≤ 10 and ≥24/care facility	23	24	17.7 Moderate dementia	Hand motor activity (finger movements, pinching, handling rubber rings, etc.)	Read aloud program	30 min, 5× week, 6 weeks	Hand movement training	NR	RBMT; Digit Span from the WMS-R; category fluency; Stop Signal task; and Attention Network Test	6 and 12 weeks	NR	−
Miu et al., (2008) [34]	85/AD, VaD, MD, UD/MMSE score ≥ 10 and ≤26/community-dwelling	36	49	18.9 Moderate dementia	Treadmill, bicycle, and arm ergometer training	Social visits and discussions on health-related topics	45–60 min, 2× week, 3 months	Only aerobic training	NR	MMSE and ADAS-Cog	3-, 6-, 9-, and 12-months post training	Good adherence (%NR)	−
Christofoletti et al., (2008) [40]	41/AD and MD// MMSE score no specified/care facility	IG1: 12 IG2: 12	17	14.0 Moderate dementia	IG1: Interdisciplinary program with strength, balance and cognition training IG2: Physiotherapy session	No motor intervention	120 min, 5× week, 6 months 60 min, 3× week, 6 months	Combined PA training	NR	MMSE and Brief Cognitive Screening Battery	6 months	NR	−
Stevens & Killeen, (2006) [33]	75/AD, UD/MMSE score ≥ 10 and ≤23/care facility	24	CG1: 30 CG2: 21	15.0 Moderate dementia	Gentile aerobic exertion of joints and large muscle groups	CG1: no intervention CG2: Social visits	30 min, 3× week, 12 weeks	Only aerobic training	Light intensity	MMSE, and CDT	12 weeks	NR	+
Van de Winckel et al., (2004) [41]	25/AD, VaD/MMSE score ≤ 23/care facility	15	10	12.0 Severe dementia	Training while sitting focus on upper and lower body strengthening, balance, trunk movements, and flexibility	Daily one-to-one conversation with therapist	30 min, daily, 3 months	Combined PA training	NR	MMSE and ADS-6	6 weeks and 3 months	NR	+ *

Intervention Group (IG); Control Group (CG); Alzheimer’s disease (AD); Mixed dementia (MD); Vascular dementia (VaD); Undefined dementia (UD); Not reported (NR); Activities of Daily Living (ADLs); Heart Rate (HR); Mini Mental state Examination (MMSE); Amsterdam Dementia screening test 6 (ADS 6); Clock-Drawing test (CDT); The Alzheimer’s Disease Assessment Scale-Cognitive Subscale (ADAS-Cog); The Stroop Colour and Word Test (SCWT); Le Rivermead Behavioural Memory Test (RBMT); Rapid Evaluation of Cognitive Function (French ERCF); The Frontal Assessment Battery (FAB); Wechsler Adult Intelligence Scale (WAIS); Wechsler Memory Scale Revised (WMS-R); Rivermead Behavioural Memory Test (RBMT); Symbol Digit Modalities Test (SDMT); Fuld Object Memory Evaluation (FOME); Verbal Fluency, Clinical Dementia Rating (CDR); Verbal fluency (VF); Severe Impairment Battery-Short From (SIB-S); Groninger Intelligence Test (GIT); Montreal Cognitive Assessment (MoCA); WHO-University of California Los Angeles-Auditory Verbal Learning test (WHO-UCLA-AVLT); Trail Making Test (TMT); Location Learning Test—Revised (LLT-R); Boston Naming Test (BNT); Hopkins Verbal Learning Test (HVLT). Effects on cognition; (+) = improvement; (*) = significant improvement; (−) = No improvement; (a) = improvement in executive functions; (b) = improvement in episodic memory; (c) = improvement in working memory; (d) = Improvement in focus and attention; (e) = improvement in visual memory; (f) = improvement in naming and abstract.

**Table 2 ijerph-18-08753-t002:** Summary measures (Mean, SD, *t*, and *p* of MMSE Scores).

	Control Group			PA Intervention				
Authors	Mean	SD	n	Mean	SD	n	*t*	*p*
Huang et al., (2019) [54]	19.47	5.73	38	21.17	5.47	36	1.304	0.196
Lamb et al., (2018) [42]	23.8	10.4	137	25.2	12.3	278	1.145	0.252
Henskens et al., (2018) [37]	9.4	5.8	16	11.6	6.5	16	1.010	0.320
Öhman et al., (2016) [48]	17.17	7.29	59	17.02	7.18	51	−0.108	0.913
Hoffmann et al., (2015) [46]	23.9	3.9	88	23.9	3.4	102	0.000	1.000
Bossers et al., (2015) [43]	15.17	4.5	36	17.16	4.33	37	1.926	0.058
Yang et al., (2015) [39]	19.54	3.43	25	22.83	2.75	25	3.742	0.000
Cheng et al., (2014) [36]	18.5	1.4	35	19.4	1.4	39	2.761	0.007
Vreugdenhil, et al., (2012) [50]	19	7.7	20	23.9	5	20	2.387	0.022
Venturelli, et al., (2011) [44]	6	2	10	12	2	11	6.866	<0.000
Kemoun et al., (2010) [49]	23.23	8.37	15	30.38	7.66	16	2.489	0.019
Eggermont, et al., (2009a) [53]	0.2	0.63	46	0.24	0.78	51	0.276	0.783
Eggermont, et al., (2009b) [52]	0.47	0.97	31	0.07	0.37	30	−2.114	0.038
Miu et al., (2008) [34]	19.2	4.2	28	17.4	5.7	24	−1.308	0.196
Christofoletti et al., (2008) [40]	14.8	1.3	17	20.2	1.6	12	10.017	<0.000
Van de Winckel, et al., (2004) [41]	11.5	5.21	9	14.4	4.4	15	1.460	0.158

**Table 3 ijerph-18-08753-t003:** Methodological quality assessment “Effective Public Health Practice Project (EPHPP) Quality Assessment Tool for Quantitative Studies”.

Study	Quality Assessment Tool for Quantitative Studies								Overall Score
	Selection Bias	Study Design	Confounders	Blinding	Data Collection Methods	Withdrawals and Dropouts	Intervention Integrity	Analyses	
Karssemeijer et al., (2019) [35]	1	1	1	2	1	1	1	1	1
Huang et al., (2019) [54]	2	1	1	2	1	1	1	1	1
Lamb et al., (2018) [42]	1	1	1	2	2	1	1	1	1
Henskens et al., (2018) [37]	2	1	1	2	1	1	1	1	1
Toots et al., (2017) [38]	1	1	1	2	1	1	1	1	1
Öhman et al., (2016) [48]	1	1	1	2	1	1	1	1	1
Kim et al., (2016) [45]	3	1	1	2	1	1	1	1	2
Cancela et al., (2016) [51]	1	1	1	2	1	1	1	1	1
Hoffmann et al., (2015) [46]	1	1	1	2	1	1	1	1	1
Bossers et al., (2015) [43]	1	1	1	2	1	1	1	1	1
Yang et al., (2015) [39]	3	2	1	3	2	3	1	1	3
Cheng et al., (2014) [36]	1	2	1	3	2	1	1	1	2
Vreugdenhil, et al., (2012) [50]	2	1	1	2	1	1	1	1	1
Venturelli, et al., (2011) [44]	3	2	1	2	1	1	1	1	2
Kemoun et al., (2010) [49]	3	2	1	2	1	1	1	1	2
Steinberg et al., (2009) [47]	3	2	1	2	1	2	1	1	2
Eggermont, et al., (2009a) [53]	2	1	1	2	1	1	1	1	1
Eggermont et al. (2009b) [52]	2	2	1	2	1	1	1	1	1
Miu et al., (2008) [34]	2	1	1	2	2	2	1	1	1
Christofoletti et al., (2008) [40]	2	1	1	2	1	3	2	1	2
Stevens & Killeen, (2006) [33]	2	1	1	3	1	3	1	1	3
Van de Winckel, et al., (2004) [41]	3	1	1	2	1	1	1	1	2

1 = Strong; 2 = Moderate; 3 = Weak.

**Table 4 ijerph-18-08753-t004:** Inclusion of methodological recommendations assessment.

Study	Implementation of Comprehensive Cognitive Measures	Implementation of Measurements Throughout the Intervention Period	Long-Term Follow-Up Measure	Target Dementia Type	Target Dementia Stage	Provide and Describe PA Characteristics of the Intervention				Total Number of Incorporated Recommendations
						Describe PA Type	Describe PA Duration	Describe PA Frequency	Describe PA Intensity	
Karssemeijer et al., (2019) [35]	+	−	(+)	−	(+)	+	+	+	+	5
Huang et al., (2019) [54]	+	(+)	−	−	+	+	+	+	−	5
Lamb et al., (2018) [42]	−	(+)	−	−	(+)	+	+	+	+	4
Henskens et al., (2018) [37]	+	(+)	−	−	−	+	+	+	(+)	4
Toots et al., (2017) [38]	(+)	−	(+)	(+)	(+)	+	+	+	+	4
Öhman et al., (2016) [48]	(+)	+	−	+	−	+	+	+	−	5
Kim et al., (2016) [45]	(+)	−	−	+	(+)	+	+	+	+	5
Cancela et al., (2016) [51]	(+)	+	−	−	−	+	+	+	(+)	4
Hoffmann et al., (2015) [46]	+	−	−	+	+	+	+	+	+	7
Bossers et al., (2015) [43]	+	−	(+)	−	(+)	+	+	(+)	+	4
Yang et al., (2015) [39]	−	−	−	−	(+)	+	+	+	+	4
Cheng et al., (2014) [36]	+	−	+	(+)	(+)	+	+	+	−	5
Vreugdenhil, et al., (2012) [50]	−	−	−	+	(+)	+	+	+	−	4
Venturelli, et al., (2011) [44]	−	−	−	+	+	+	+	+	−	5
Kemoun et al., (2010) [49]	−	−	−	+	−	+	+	+	+	5
Steinberg et al., (2009) [47]	+	(+)	−	+	(+)	+	+	−	−	4
Eggermont, et al., (2009a) [53]	+	−	(+)	−	(+)	+	+	+	−	4
Eggermont et al. (2009b) [52]	(+)	−	(+)	−	(+)	+	+	+	−	3
Miu et al., (2008) [34]	−	−	+	−	(+)	+	+	+	−	4
Christofoletti et al., (2008) [40]	−	−	−	−	(+)	+	+	+	−	3
Stevens & Killeen, (2006) [33]	(+)	−	−	−	(+)	+	+	+	−	3
Van de Winckel, et al., (2004) [41]	−	(+)	−	−	−	+	+	+	−	3

+ Fully incorporated; (+) partly incorporated; − Not incorporated.

## Data Availability

The datasets used and/or analysed during the current study are available from the corresponding author on reasonable request.

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
