# Peer review of "Physical Activity Interventions and Their Effects on Cognitive Function in People with Dementia: A Systematic Review and Meta-Analysis"

_ijerph, 2021, doi:10.3390/ijerph18168753_

Round 1
Reviewer 1 Report
This is a systematic review and meta-analysis of RCTs on the effects of physical activity on cognition among dementia patients. The rationale is unclear and overall the writing would benefit from a copy editor.
Abstract:
The Conclusions section is poorly worded. A copy editor may be beneficial.
Intro:
Rather than focusing on previous literature on effects of exercise among dementia patients, excessive and unnecessary detail is provided on topics such as the definition of aging and the definition of physical activity. Why did the authors choose to focus on dementia patients rather than participants with mild cognitive impairment? The rationale of focusing on exercise among dementia patients should be more clearly stated.
Objective:
A research objective should not be in the form of a question.
Materials and Methods:
Dementia is a clinical diagnosis. It is unclear why strict cut-offs on MMSE, MoCA, etc. were used as part of inclusion criteria, considering that norms vary drastically by age and years of education. How was dementia diagnosed – in what clinical settings and by whom? What do the authors mean by: “The primary outcome involved individuals with dementia and addressed the cognitive function”?
Why were pilot RCTs excluded?
What proportion of corresponding authors contacted did not respond to requests for additional data?
Discussion:
The most important clinical consideration for interpretation of results is PA engagement and adherence to programs, considering the study population consists of individuals with dementia. As dementia progresses (the patient loses ability to comprehend/participate in such activities, becomes bed-bound, mute, etc.), engagement and adherence will naturally decline. Due to this, the clinical relevance is limited, particularly if cognition is assessed as the outcome of interest, rather than behavioral or functional measures, which often are more relevant for the patient’s quality of life in later stages of dementia.
The implications for future research and policy noted by the authors would appear to be premature, given the limited conclusions based on this systematic review and meta-analysis.
Conclusions:
Publication bias is a significant limitation. Given the heterogeneity of studies, it is unclear whether the meta-analysis provides useful conclusions. Clinically, the question of whether physical activity is beneficial among dementia patients is most relevant for individuals for mild dementia. The impact of physical activity on cognitive outcomes of individuals with mild cognitive impairment likely holds greater clinical relevance and public health significance.
Author Response
Manuscript ID: ijerph-1303754
Physical Activity interventions and their effects on Cognitive Function in People with dementia: A systematic review
Maria Isabel Cardona, MA; Adel Afi; Nemanja Lakicevic; Jochen Rene Thyrian
International Journal of Environmental Research and Public Health
Dear Editor-in-Chief,
Thank you for the reviewers' comments regarding the manuscript "Physical Activity Interventions and their Effects on Cognitive Function in People with dementia: A systematic review". We have carefully reviewed the comments and have revised the manuscript accordingly. Changes are shown in red and we have included the certificate of professional editing with this revision.
We believe that the manuscript improved, addressing the reviewers and now conveys the relevant information properly.
Reviewer 1.
Thank you for your review of our paper. Please find our point-to point reply below.
1) The Conclusions section is poorly worded. A copy editor may be beneficial.
Answer:
Thank you for your recommendation. A professional English editing service as suggested revised and proofread the manuscript. We have included the certificate of professional editing with this revision.
Intro:
2) Rather than focusing on previous literature on effects of exercise among dementia patients, excessive and unnecessary detail is provided on topics such as the definition of aging and the definition of physical activity.
Answer:
Thank you for your remark. We agree with the reviewer that there was unnecessarily detailed information regarding the definition of ageing. Therefore, we have shortened this information.
Physical activity (PA) is used in the included studies as an umbrella term for exercise and training. Therefore, we believe that it is essential to explain what kind of definitions are used to describe PA in a more elaborately way. Moreover, we addressed belonging intensities since, in our opinion, this is relevant as we described in the manuscript different intensities in our result table.
We believe that manuscript focused on previous literature regarding the effects of exercise among dementia patients, including results from the latest Cochrane review by Forbes et al., 2015, which addresses the effects of exercise interventions in patients who have dementia on cognitive outcomes as the primary outcome. This is crucial work in this field of research and healthcare.
3) Why did the authors choose to focus on dementia patients rather than participants with mild cognitive impairment?
Answer:
We decided to focus on patients with dementia rather than participants with mild cognitive impairment (MCI) since not all MCI patients will develop dementia. Furthermore, patients with dementia, a specific disease, will continue to decrease memory and functioning over time significantly. Thus, the effects of PA may also be different considering that they will undergo a progressive cognitive decline over time.
4) The rationale of focusing on exercise among dementia patients should be more clearly stated.
Answer
Thank you for your suggestion. In order to provide a more clear rationale on exercise among dementia patients, we modified the manuscript as follows:
- 89 “Although the positive effects of exercise on cognition in older adults have been researched, the influence of PA on cognitive function of PwD are still not well understood [15]” (…).
Objective:
5) A research objective should not be in the form of a question.
Answer
We modified the research objectives as follow:
- 121 “Primary objective
To identify physical activity interventions effects on cognitive function in persons diagnosed with dementia compared to the control group
Secondary objective
To recognize if recent physical activity interventions address methodological barriers reported in previous reviews and provide clearer conclusions about PA effects on cognition of PwD.
Materials and Methods:
6) Dementia is a clinical diagnosis. It is unclear why strict cut-offs on MMSE, MoCA, etc. were used as part of inclusion criteria, considering that norms vary drastically by age and years of education.
Answer: Thank you very much for your valuable comment. We reported in the inclusion criteria the most conventional Cut-off scores, as change values alongside the spectrum of cognitive impairment, to assure the inclusion of persons diagnosed with dementia.
However, to prevent imputing non-realistic values and not discard any study that considered different cut-off values bearing in mind educational level and age, each study, one by one, was screened to verify which cut-off point was used.
7) How was dementia diagnosed – in what clinical settings and by whom?
Answer:
Thank you for your interesting remark. We did not extract specific information on the clinical setting and the personnel in charge of the diagnosis. In line with our objectives, we prioritized identifying and reporting information more focused on which cognitive assessment was used (see table 1); participant’s clinical diagnosis (e.g. MMSE score, dementia severity, dementia type); and recruiting settings.
8) What do the authors mean by: “The primary outcome involved individuals with dementia and addressed the cognitive function”?
Answer:
Thank you for your observation. We also believe that phrase is unclear and unnecessary. Thus, we deleted it from the manuscript.
9) Why were pilot RCTs excluded?
Answer:
We excluded pilot RCTs since it was hoped to include more rigorously planned and informed randomised controlled trials.
10) What proportion of corresponding authors contacted did not respond to requests for additional data?
Answer:
Twenty-two authors were contacted, of which six did not respond to the request for additional data.
Discussion:
11) The most important clinical consideration for interpretation of results is PA engagement and adherence to programs, considering the study population consists of individuals with dementia. As dementia progresses (the patient loses ability to comprehend/participate in such activities, becomes bed-bound, mute, etc.), engagement and adherence will naturally decline. Due to this, the clinical relevance is limited, particularly if cognition is assessed as the outcome of interest, rather than behavioral or functional measures, which often are more relevant for the patient’s quality of life in later stages of dementia.
Answer:
Although still under debate, there is some evidence that PA leads to improved cognitive function by stimulating neurotrophic factors. However, particularly in people with dementia, there is very little research in this regard. Therefore, it is crucial to continue generating scientific evidence to support claims such as those made by the evaluator: “the clinical relevance is limited, particularly if cognition is assessed as the outcome of interest, rather than behavioural or functional measures, which often are more relevant for the patient’s quality of life in later stages of dementia”.
12) The implications for future research and policy noted by the authors would appear to be premature, given the limited conclusions based on this systematic review and meta-analysis.
Answer:
We agree with the reviewer, therefore, we added the following information to the manuscript:
- 410 “Based on the findings from the most recent studies exploring PA effects on cognition of PwD, these might be some implications to consider for future research and policy. However, it is essential to take them with caution, as the reported studies still present certain limitations”.
Conclusions:
13) Publication bias is a significant limitation. Given the heterogeneity of studies, it is unclear whether the meta-analysis provides useful conclusions. Clinically, the question of whether physical activity is beneficial among dementia patients is most relevant for individuals for mild dementia. The impact of physical activity on cognitive outcomes of individuals with mild cognitive impairment likely holds greater clinical relevance and public health significance.
Answer:
Thank you for your comment. Although a high percentage of heterogeneity was presented, we still believe that the present study provides some valuable conclusions. The authors believe that the systematic review provides an update of recent work on this topic. Therefore, it enables us to see recent methodological improvement (e.g. larger samples and maintained higher levels of follow-up). Hence, robust studies may provide more consistent conclusions to the limited knowledge we currently have on the effects of physical activity on cognitive functioning in people with dementia. For example, the critical role that PA dose-response may play in effects on cognition in this population.
Likewise, as the evaluator mentions, this study demonstrates the need for future studies to focus on individual characteristics of the participants, such as the severity of dementia, as there is limited available evidence to determine at what stage of dementia physical activity would be of most benefit.

Reviewer 2 Report
It was a pleasure to review this paper. The entitled paper “Physical Activity Interventions and Their Effects on Cognitive Function in People with Dementia: A Systematic Review” by Maria Cardona, Adel Afia, Nemanja Lakicevic and Jochen R Thyriana aims to identify and summarize PA interventions and their effects on cognitive function among persons with dementia.
Although past reviews had addressed this topic, recently, an increased amount of RCTs in this area has been published. Therefore, this manuscript represents an essential update, on a very important topic. In general, this manuscript is well written and well-structured. The methodological procedure is adequate for a systematic review and meta-analysis.
However, I would suggest that the authors address the following comments, mainly focused on the introduction:
1) after the title, the authors names are marked with abc and the corresponding institutions are with 123.
2) Line 13, page 1, I believe that starting the sentence straight away with “Physical activity(...)” would it be better. Same with line 21.
3) Line 23, page1, it wasn´t quite clear this sentence means: “Our findings suggest that there is an increase in studies methodological aspects”.
4) Line 39/40 the authors refer that “Proper CNS functioning is crucial in the light of dementia disease, an umbrella term for multiple neurodegenerative diseases”. This information doesn't seem to be correct. When neurodegenerative diseases are installed, CNS damages have already occurred.
5) Between line 49 and 55, the authors refer to the modifiable risk factors, and although I agree that this idea this should be highlighted, I think the authors should also address non-modifiable risk factors. Even because they represent most of the risk factors.
6) Line 60/61, the following sentence “These processes compensate for losses of brain tissue while the brain is ageing [11].” seems to be telling us that cognitive reserve compensates (completely) the losses of brain issues. I believe it would be beneficial to use the term “attenuate” instead of “compensate”.
7) In a more general perspective, the introduction would benefit if the authors could address slightly the mechanisms behind dementia and if the authors could address other processes (not just cognitive reserve) triggered by the physical activity that may be beneficial for cognition. This could be also addressed in the discussion section.
8) In the methods section, line 122, states “including strength, aerobic or cognitive training, to improve(..)”. The authors meant, “It is aerobic or cognitive training?”
9) line 126/7, specific cut-off points for MMSE and MOCA are used. This review integrates 22 RCT of different countries, and do all these countries have the same cut off points?
Author Response
Manuscript ID: ijerph-1303754
Physical Activity interventions and their effects on Cognitive Function in People with dementia: A systematic review
Maria Isabel Cardona, MA; Adel Afi; Nemanja Lakicevic; Jochen Rene Thyrian
International Journal of Environmental Research and Public Health
Dear Editor-in-Chief,
Thank you for the reviewers' comments regarding the manuscript "Physical Activity Interventions and their Effects on Cognitive Function in People with dementia: A systematic review". We have carefully reviewed the comments and have revised the manuscript accordingly. Changes are shown in red and we have included the certificate of professional editing with this revision
We believe that the manuscript improved, addressing the reviewers and now conveys the relevant information properly.
Reviewer 2.
Thank you for your review of our paper. Please find our point-to point reply below.
I would suggest that the authors address the following comments, mainly focused on the introduction:
1) After the title, the authors names are marked with abc and the corresponding institutions are with 123.
Answer:
Thank you for this important remark. We marked with abc the corresponding institutions as well.
2) Line 13, page 1, I believe that starting the sentence straight away with “Physical activity(...)” would it be better. Same with line 21.
Answer:
Thank you very much for your suggestion. Both sentences were modified as follow:
- 15 “Physical activity (PA) emerges as an alternative non-pharmacological approach (…)”
- 23 “Twenty-two PA interventions met the eligibility criteria and showed a medium effect size on the cognitive function of PwD (…)”.
3) Line 23, page1, it wasn´t quite clear this sentence means: “Our findings suggest that there is an increase in studies methodological aspects”.
Answer:
We agree with the reviewer that the sentence was not entirely clear. Therefore we remove it from the abstract.
4) Line 39/40 the authors refer that “Proper CNS functioning is crucial in the light of dementia disease, an umbrella term for multiple neurodegenerative diseases”. This information doesn't seem to be correct. When neurodegenerative diseases are installed, CNS damages have already occurred.
Answer:
Thank you for this relevant remark. We modified the statement in the manuscript as follows:
L.41 “The physiological characteristics of dementia, an umbrella term for multiple neurodegenerative diseases [5], has been linked to severe degeneration of brain cells and synapses in certain areas of the CNS, including the temporal, parietal and frontal cortices [6]. Damage in these areas manifests itself through memory and learning deficits [6]. In addition, dementia affects emotional regulation, social functioning, and activities of daily living [5]”.
5) Between line 49 and 55, the authors refer to the modifiable risk factors, and although I agree that this idea this should be highlighted, I think the authors should also address non-modifiable risk factors. Even because they represent most of the risk factors.
Answer:
Thank you for your valuable remark. In the following manuscript line, we included non-modifiable risk factors. However, we did not expand much further into these factors, as we intended to emphasize the modifiable factors that have been identified in the literature as being associated with physical activity.
- 55 “Other non-modifiable factors linked to dementia include age, sex, inflammation, and comorbidity, and genetic, environmental, and lifestyle factors [10]”.
6) Line 60/61, the following sentence “These processes compensate for losses of brain tissue while the brain is ageing [11].” seems to be telling us that cognitive reserve compensates (completely) the losses of brain issues. I believe it would be beneficial to use the term “attenuate” instead of “compensate”.
Answer:
We used the suggested term as follow:
L.68 “These processes attenuate for losses of brain tissue while the brain is aging [14]
7) In a more general perspective, the introduction would benefit if the authors could address slightly the mechanisms behind dementia and if the authors could address other processes (not just cognitive reserve) triggered by the physical activity that may be beneficial for cognition. This could be also addressed in the discussion section.
Answer
Thank you for your suggestion, we addressed the mechanisms behind dementia as follows:
- 50 “The causes of dementia onset are not fully understood, but notably, the mechanism underlying dementia is associated with abnormal protein deposits that coexist with neurovasculature at different stages of the disease, which affect brain functioning [9]. Depending on the type of dementia, different protein accumulations are observed. For instance, alpha-synuclein protein is linked to Lewy body dementia, whereas beta-amyloid and tau proteins are both related to Alzheimer's disease, the most common form of dementia. Inadequate blood flow can lead to vascular dementia [9]”.
Also, we shortly addressed other processes triggered by physical activity in the introduction as is shown in the following line:
- 64 “This is mainly because regular PA improves the strength of cells and tissues to respond to oxidative stress, vascularization, and energy metabolism (…)”.
- 71 Moreover, the positive effects of PA on cognition appear to be influenced by preventing cardiovascular risk factors (e.g. obesity, hypertension, diabetes) which at the same time are linked with greater probability of dementia progression [16]. Additionally, neuroimaging methods add further evidence of PA impact on brain activity on cognitive function [16]. For instance, an enlarged level of connection was detected between the default mode network (DMN), which is a control structure widely known to be responsible for introspection and memory retrieval, after PA training [17]. Precisely, animal models of Alzheimer's disease (AD) illustrate that PA is an effective way to positively modify pathophysiological processes, including β-amyloid (Aβ) burden, tau phosphorylation, and neuronal loss [18].
8) In the methods section, line 122, states “including strength, aerobic or cognitive training, to improve(..)”. The authors meant, “It is aerobic or cognitive training?”
Answer
For more clarity in that statement, we modified that line in the manuscript as follow:
- 133 “The exercise group required implementing a PA programme, including strength, aerobic, and balance exercises, as well as interventions combining physical and cognitive exercises for improving cognition performance in PwD.”
9) line 126/7, specific cut-off points for MMSE and MOCA are used. This review integrates 22 RCT of different countries, and do all these countries have the same cut off points?
Answer:
Thank you for this valuable remark. As you and the literature mentioned, the cut-off points should be stratified by race/ethnicity and education to guarantee accurate detection of dementia. However, the studies included do not mention further stratification. For example, some studies used a conventional cut-off of 26 points (for Moca) and 23 points (for MMSE) as the ideal level to detect dementia. However, your comment is precious and evidences the importance of reassessing the cut-off points to evaluate minority populations.

Reviewer 3 Report
Thank you for the opportunity to revise this systematic review.
The paper is interesting, and it addressed to an important and debatable argument.
The authors followed the PRISMA guidelines and focused on RCT studies involving participants with dementia.
Honestly, i didn't find particular criticisms: All the main points indicated by PRISMA guidelines were reported in the manuscript; the analysis seems appropriate and clearly explained; discussion seems to be pertinent.
The main problem of this paper is that, as the authors themselves reported, "It is possible to say that these results continue to resemble those obtained in other reviews since we are talking about the twenty-two included RCTs applied from the year 2004 to the present." In fact, the present manuscript reported the same conclusions and the same limitations of previous similar systematic reviews. The recent RCTs (published in the last 2-3 years) seem to not provide new information to discuss.
However, as previously stated, the overall quality of the paper is good and, in my opinion, the paper is suitable for publication.
Author Response
Manuscript ID: ijerph-1303754
Physical Activity interventions and their effects on Cognitive Function in People with dementia: A systematic review
Maria Isabel Cardona, MA; Adel Afi; Nemanja Lakicevic; Jochen Rene Thyrian
International Journal of Environmental Research and Public Health
Dear Editor-in-Chief,
Thank you for the reviewers' comments regarding the manuscript "Physical Activity Interventions and their Effects on Cognitive Function in People with dementia: A systematic review". We have carefully reviewed the comments and have revised the manuscript accordingly. Changes are shown in red and we have included the certificate of professional editing with this revision
We believe that the manuscript improved, addressing the reviewers and now conveys the relevant information properly.
Reviewer 3
Thank you for your review of our paper. Please find our point-to point reply below.
The main problem of this paper is that, as the authors themselves reported, "It is possible to say that these results continue to resemble those obtained in other reviews since we are talking about the twenty-two included RCTs applied from the year 2004 to the present." In fact, the present manuscript reported the same conclusions and the same limitations of previous similar systematic reviews. The recent RCTs (published in the last 2-3 years) seem to not provide new information to discuss.
Answer:
Thank you for your valuable remark, it allowed us to notice that in the writing it was not clear the innovative aspects of our study. Therefore, the following sentence was rewritten to emphasize our contribution even though there are still similarities with previous revisions:
- 330 “Even though these results resemble those obtained by Forbes et al. [22], if we look at the newly added trials implemented in recent years, a slight difference is revealed. In particular, six recent trials [32, 35, 39, 40, 43, 45] showed more powerful designs since they included larger sample sizes. In this way, this finding enables us to see small developments and progress in this particular research field, including more solid methodological designs and higher statistical power in the most recent studies. Therefore, the results presented in these trials might lead to more precise conclusions about the effects of PA on the cognition of PwD”.
